# VVX001 Induces preS-Specific Antibodies Reacting to Common HBV Genotypes in Hepatitis B Virus (HBV) Carrier Mice

**DOI:** 10.3390/vaccines13080854

**Published:** 2025-08-12

**Authors:** Inna Tulaeva, Maryline Bourgine, Carolin Cornelius-Nikl, Alexander Karaulov, Rainer Henning, Marie-Louise Michel, Rudolf Valenta

**Affiliations:** 1Institute of Pathophysiology and Allergy Research, Center for Pathophysiology, Infectiology and Immunology, Medical University of Vienna, 1090 Vienna, Austria; inna.tulaeva@meduniwien.ac.at (I.T.);; 2Laboratory of Immunopathology, Department of Clinical Immunology and Allergology, I. M. Sechenov First Moscow State Medical University, 119991 Moscow, Russia; 3Life Improvement by Future Technologies (LIFT) Center, 115478 Moscow, Russia; 4Virology Department, Institut Pasteur, Université Paris Cité, 75015 Paris, France; maryline.bourgine@pasteur.fr (M.B.);; 5Viravaxx AG, 1190 Vienna, Austria; 6Center for Molecular Allergology, Karl Landsteiner University of Health Sciences, 3500 Krems, Austria

**Keywords:** hepatitis B, preS, vaccination, grass pollen allergy vaccine BM32, VVX001, viral vector, antibody response, epitopes, genotype cross-reactivity

## Abstract

*Background*: Chronic hepatitis B (CHB) remains being a major public health threat, and currently existing CHB therapies have limited efficacy and side effects. We have recently developed a vaccine termed VVX001 based on a recombinant fusion protein consisting of the preS domain of the large surface protein of hepatitis B virus (HBV) fused to grass pollen allergen peptides. VVX001 has been shown to induce preS-specific antibodies in grass pollen allergic patients, and sera of immunized subjects inhibited HBV infection in vitro. *Methods*: In this study we investigated if immunization with VVX001 can induce preS-specific antibodies in CHB using the adeno-associated virus (AAV)-HBV murine model of CHB. Six groups of C57BL/6 female mice (n = 6) were transduced with AAV-HBV or AAV-Empty, and after six weeks, they were immunized five times with 20 µg of aluminum hydroxide-adsorbed VVX001 or preS or vehicle (Alum alone). Serum samples were taken continuously. Two weeks after the last immunization, spleen and liver mononuclear cells were collected. Serum reactivity to preS and preS-derived peptides was assessed by ELISA. B-cell responses were measured by ELISPOT assay, and intrahepatic lymphocyte (ILH) counts were determined by FACS. HBV DNA, HBsAg, HBeAg, ALT, and AST were assessed using commercial kits. *Results*: Our results show that VVX001 induces preS-specific IgG antibodies that cross-react with different HBV genotypes A-H and are directed against the sodium taurocholate co-transporting polypeptide (NTCP) receptor binding site of preS both in mice with and without HBV. Actively immunized AAV-HBV-treated mice had a higher number of intrahepatic lymphocytes than vehicle-vaccinated and mock-transduced animals. *Conclusions*: These findings encourage performing further trials to study the potential of VVX001 for therapeutic vaccination against CHB.

## 1. Introduction

Hepatitis B virus (HBV) is a global health problem affecting more than 250 million people who are chronically infected [1]. Despite the availability of effective prophylactic vaccines, chronic hepatitis B (CHB) infection still resulted in more than 800,000 deaths a year due to HBV-associated liver failure or hepatocellular carcinoma [2,3]. It has been well documented that approximately 90% of the infected adult patients are able to mount a robust immune response that is sufficient to control HBV infection; in addition, each year about 2–3% of CHB patients can spontaneously seroconvert to anti-HBs positive subjects and control their infection [3]. Current therapies against CHB infection include potent direct-acting antivirals and pegylated interferon (IFN) alpha. However, there is the risk of viral resistance, drug toxicity, and side effects. Furthermore, these therapies rarely achieve functional cure as defined by hepatitis B surface antigen (HBsAg) loss and anti-HBs seroconversion [4]. These facts encourage the development of immunotherapies for the treatment of CHB. Key elements for controlling HBV infection are the induction of neutralizing antibodies against HBV, further leading to its clearance, and the induction of a broad and robust antigen-specific T cell response. However, the ongoing HBV replication together with the high concentrations of circulating HBsAg in CHB patients causes a general hypo-responsiveness of the immune system to the persisting virus [5]. In these patients, HBV-specific T cells are deleted or functionally exhausted most likely due to the repeated exposure to a high antigen load. Likewise, it has been shown that therapeutic vaccine-induced antibody responses are inhibited due to the presence of persistently high HBsAg levels [6]. Therefore, the major objective of a therapeutic vaccination against CHB is to restore the functionality of antiviral B-cell and T-cell responses.

The HBsAg gene includes preS1, preS2, and S domains. Its translation may result in formation of the large (LHBs), the medium (MHBs) HBsAg proteins, containing preS1 + preS2 + S domains or preS2 + S, respectively, or the small (SHBs) surface protein only containing the S domain [1].

Therapeutic vaccination for CHB had not yet demonstrated clear efficacy in clinical trials, which could be attributed to suboptimal vaccine design based mainly on SHBs [7]. The main obstacle in CHB is the large quantities of HBsAg forming subviral particles (SVPs) mostly composed of SHBs, which has been proposed to cause antigen-specific immune tolerance. Alternative HBV vaccine candidates other than exclusively SHBs-based are eventually promising therapeutic vaccines [8]. The preS1 domain serves as a potential target for HBV vaccination because of its key role in hepatocyte entry via the sodium taurocholate co-transporting polypeptide (NTCP) receptor [9] and in the assembly and release of HBV virions [10]. More importantly, preS1 exists primarily in mature infectious HBV virions, and the antigen availability is much lower than that of HBsAg [11], thus having the potential to overcome immune tolerance to HBV. Inhibition of the preS-NTCP interaction is likely to efficiently prevent re-infection and block de novo infections of naïve hepatocytes or those that arise during hepatocyte turnover. Therefore, a blockade of the preS-NTCP interaction may reduce the number of infected cells while accelerating the process of virus elimination.

When used in allergy vaccines as an immunological carrier, preS in combination with allergen-derived peptides was shown to be highly immunogenic [12]. One of the recombinant peptide carrier vaccines, VVX001, is composed of the preS (preS1 + preS2) of the LHBs fused to peptide sequences from the grass pollen allergen Phl p 5 at both ends [13]. We have previously shown that by vaccination with VVX001, a potent IgG response to preS could be elicited even in non-responders to HBsAg-based prophylactic vaccines [14]. The antibody response is strongly focused on the sequence motif in the preS1 domain responsible for HBV entry in hepatocytes [15]; moreover, sera of subjects immunized with VVX001 efficiently inhibited HBV infection in vitro [12]. Furthermore, escape mutations seem to be less problematic for preS as compared to the HBsAg [5,16,17].

These findings encouraged us to study the potential of the preS-based vaccine VVX001 in reversing immune tolerance in CHB in an animal model. Chimpanzees are the only nonhuman immune-competent animals that are naturally susceptible to chronic HBV infection, but this model is difficult to access, and there are ethical considerations. Mice have well-characterized immune systems and are widely used in research, but they are naturally not susceptible to HBV. Therefore, various strategies have been exploited to introduce the HBV genome into the hepatocytes of mice. One of the delivery methods is to introduce a replication-competent HBV genome into the mouse liver by hydrodynamic injection through the tail vein; even though HBV replicates in the mouse liver, the virus is cleared rapidly by the immune responses against HBV proteins [18]. An alternative method uses an adenoviral or adeno-associated viral vector to transfer copies of the HBV genome into immunocompetent mice. Depending on the dose of the vector injected, acute or chronic HBV infection can be induced [19,20]. Here we used the chronic HBV infection murine model established by transducing the liver with an adeno-associated virus (AAV) serotype 2/8 carrying a replication-competent HBV-DNA genome which resulted in a long-term chronic HBV-carrier state [21]. The aim of this study was to evaluate if immunization with VVX001 or preS can induce an HBV-specific immune response in the AAV-HBV mouse model of CHB.

## 2. Materials and Methods

### 2.1. Animal Experiments

C57BL/6 female mice were purchased from Janvier Labs (Le Genest-Saint-Isle, France). Mice were housed under pathogen-free conditions and used at 6–8 weeks of age. The mice were housed in groups and provided with appropriate bedding. They were subjected to standardized light–dark cycles and given ad libitum access to food and water. The scheme of the experiment is depicted in Figure 1. After a one-week acclimatization period, the livers of mice were transduced with either an AAV serotype 2/8 carrying a replication-competent HBV DNA genome (3 groups of 6 mice in each) or an AAV-Empty vector (3 groups of 6 mice in each). The sample size of six mice per group was determined based on the results of our previous study in order to achieve statistically significant differences between the groups, particularly in the case of partial control [12]. The HBV genome was introduced by i.v. injection with 5 × 10^10^ viral genome equivalents (vg) of AAV2/8-HBV in the tail vein as described [21]. Serum samples were obtained from mice at baseline, shortly before the first immunization, on the days of the second to fifth injection and two weeks after the last injection (Figure 1). Five immunizations with three week intervals were performed by subcutaneous (s.c.) injections of 20 μg of VVX001 or preS adsorbed onto aluminum hydroxide under the scruff of the neck without anesthesia. Control mice received an identical volume of aluminum hydroxide (vehicle, 200 μL). Following the AAV injections and vaccination, animals were monitored at least three times per week for any signs of discomfort. Mice exhibiting any of the following humane endpoints were excluded from the analysis: progressive loss of activity; abnormal posture; signs of infection; or weight loss of more than 15%. All protocols have been reviewed and approved by the institutional animal care committee of Institute Pasteur for compliance with the French and European regulations on animal welfare and with public health service recommendations (authorization number No. 02651.02).

### 2.2. Vectors and Immunogens

An AAV serotype 2/8, either carrying a replication-competent HBV DNA genome (AAV-HBV) or not (AAV-Empty), were used. The AAV-HBV (batch GVPN #6565) and AAV-Empty (batch GVPN #5198) vectors were stored at −80 °C at a concentration of 1.2 × 10^13^ and 1.0 × 10^13^ viral genomes (vg)/mL, respectively. Before use, AAV vectors were thawed at room temperature and then diluted in sterile phosphate-buffered saline (PBS) to reach a titer of 5 × 10^11^ vg/mL. Mice were injected intravenously (i.v.) with 100 μL of this solution (dose/mouse: 5 × 10^10^ vg) (Figure 1).

VVX001 is a recombinant fusion protein composed of the preS (preS1 + preS2, GenBank: AAT28735) domain of the large surface antigen (LHBs) of HBV genotype A2 and peptides derived from the grass pollen allergen Phl p 5 at both the C- and N-terminus. VVX001 is one of the components of the grass pollen allergy vaccine BM32, namely BM325; a detailed description of the structure and peptide sequences has been reported [13]. As an immunogen of comparison, preS (preS + preS2, genotype A2, GenBank: AAT28735) alone containing a C-terminal hexahistidine tag was expressed in *E. coli* and purified as described [12].

### 2.3. Preparation of Cells

Spleen and liver mononuclear cells as well as blood samples from each mouse were collected 2 weeks after the last immunization and depleted of red blood cells using Lysing Buffer (BD Biosciences, Franklin Lakes, NJ, USA, Cat. 555899). The liver mononuclear cells underwent a specific preparation as described below and according to a method previously published by Tupin et al. [22] with minor modifications. After mouse euthanasia, the liver was perfused with 10 mL of sterile PBS via the hepatic portal vein using a syringe with a G25 needle until the organ was becoming pale, then it was harvested in Hank’s Balanced Salt Solution (HBSS) (Gibco, Thermo Fisher Scientific, Waltham, MA, USA, Cat. 24020) +5% decomplemented fetal calf serum (FCS). Afterward, the liver was gently pressed through a 100 μm cell strainer (BD Falcon, Franklin Lakes, NJ, USA, Cat. 352360), and cells were suspended in 30 mL of HBSS + 5% FCS. Cell suspension was centrifuged at 50× *g* for 5 min, and then the supernatants were centrifuged at 289× *g* for 10 min at 4 °C. After centrifugation, supernatants were discarded and pellets were re-suspended in 15 mL of 35% isotonic Percoll solution (GE Healthcare, Chicago, IL, USA, Cat. #17-0891-01) diluted into RPMI 1640 (Gibco, Thermo Fisher Scientific, Cat. 31870) at room temperature and transferred in 15 mL tubes. Cells were further centrifuged at 1360× *g* for 25 min at room temperature. Afterward, the supernatant was discarded by aspiration and the pellet containing mononuclear cells was washed twice with HBSS + 5% FCS. Cells were then resuspended in complete medium (α-minimal essential medium (Gibco, Thermo Fisher Scientific, Cat. 22571) supplemented with 10% FCS (Hyclone, Thermo Fisher Scientific, Cat. SH30066), 100 U/mL penicillin + 100 μg/mL streptomycin + 0.3 mg/mL L-glutamine (Gibco, Thermo Fisher Scientific, Cat. 10378), 1X non-essential amino acids (Gibco, Thermo Fisher Scientific, Cat. 11140), 10 mM Hepes (Gibco, Thermo Fisher Scientific, Cat. 15630), 1 mM sodium pyruvate (Gibco, Thermo Fisher Scientific, Cat. 11360) and 50 μM β-mercaptoethanol (LKB, Piscataway, NJ, USA, Cat. 1830).

### 2.4. B Cell Enzyme-Linked Immunospot (ELISPOT) Assays

Antibody-producing B cells amongst splenocytes were quantified by ELISPOT assay after antigen stimulation, as previously described. To determine the number of antibody-secreting cells (ASC), sterile MSIP 96-well plates (Millipore, Bedford, MA, USA) were pre-wetted for 1 min with 15 μL 35% ethanol, washed with water, coated with MHBs (preS2 + S, genotype D), HBcAg (genotype A) or preS, and incubated overnight at 4 °C. Wells of ELISPOT plates were also coated with an anti-mouse IgG coating antibody that binds to the antibodies released by the antibody-secreting plasma cells as positive controls, and keyhole limpet hemocyanin (KLH) was used as a negative control to determine background levels. Wells were post-coated with a blocking agent for 2 h at 37 °C and subsequently incubated with cell preparations (5 × 10^5^ cells/well). After an overnight incubation, cells were washed away, and the antibodies captured by the immobilized antigen or antibody were detected with biotinylated anti-mouse IgG antibodies followed by incubation with a phosphatase alkaline-labeled detection antibody. Revelation was performed by adding the BCIP/NBT substrate. The response to the stimulant was considered positive if the median number of antibody-secreting cells (ASC) in triplicate wells was at least twice as much as observed in control wells stimulated by KLH and when at least 10 antibody-secreting cells per million splenocytes were detected after subtraction of the background.

### 2.5. Cell Labeling and Intracellular Cytokine Staining (ICS)

Due to the limited number of lymphocytes collected from the liver, all cells were used for the experiment and divided according to the number of tested parameters. As a consequence, the number of cells/well in the same experiment is identical for the same mouse but variable among different mice.

The detection of cell populations was performed by surface labeling of purified liver mononuclear cells: the cells were seeded in U-bottom 96-well plates and washed with PBS FACS (PBS containing 1% bovine serum albumin and 0.01% sodium azide). Cells were then incubated with 5 μL of PBS FACS containing a rat anti-mouse CD16/CD32 antibody and a viability marker (LD fixable yellow, Thermo Fisher Scientific, Cat. L34959) for 10 min in the dark at 4 °C. Then, cells were stained for 20 min in the dark at 4 °C with 25 μL of PBS FACS containing a mix of monoclonal antibodies (Mab). The mix was composed of hamster Mab anti-mouse CD3-PEVio770 (Miltenyi, Bergisch Gladbach, Germany, Cat. 130-102-359), CD8 (rat Mab anti-mouse CD8-APC-H7, BD Biosciences, Cat. 560182), rat Mab anti-mouse CD4-AF647 (BD Biosciences, Cat. 557681), rat Mab anti-mouse PE-Cy5.5 (eBioscience, Thermo Fisher Scientific, Cat. 35-0193-80), rat Mab anti-mouse NK P46 (BioLegend, San Diego, CA, USA, Cat. 137612) and rat Mab anti-mouse F4/80 FITC (BioLegend, Cat. 123108). After washes, cells were fixed in PBS FACS containing 1% formaldehyde, washed, and resuspended in PBS FACS. Afterward, analysis by flow cytometry using an Attune (ThermoFisher, Waltham, MA, USA) analyzer was performed.

ICS assays were conducted on both splenocytes and liver mononuclear cells. Cells were seeded in U-bottom 96-well plates which were then incubated overnight at 37 °C either in complete medium alone as a negative control or in 3 pools of peptides from the HBc, MHBs, and preS proteins at a concentration of 2 µg/mL. After one hour of incubation, Brefeldin A at 2 µg/mL (B6542, Sigma-Aldrich, Burlington, MA, USA) was added to the cells. Cells were then cultured overnight, then washed with PBS FACS and incubated with 5 µL of PBS FACS containing rat anti-mouse CD16/CD32 antibody and a viability marker (Zombie Violet Fixable Viability Kit, BioLegend, BLE423114) for 10 min in the dark at 4 °C. Afterward, cells were stained for 20 min in the dark at 4 °C with 25 µL of PBS FACS containing a mix of Mab. The mix was composed of hamster Mab anti-mouse CD3-PerCP-Vio700 (Miltenyi, 130-119-656), rat Mab anti-mouse CD8-APCH7 (BD Biosciences, 560182), rat Mab anti-mouse CD4-PE-Cy7 (BD Biosciences, 552775). After several washes, cells were fixed and permeabilized for 20 min in the dark at room temperature with Cytofix/Cytoperm and washed with Perm/Wash solution (BD Biosciences, 554714) at 4 °C. ICS with antibodies against IFNγ (rat Mab anti-mouse IFN-APC, clone XMG1.2, BD Biosciences, 554413), interleukin (IL)-2 (rat Mab anti-mouse IL2-PE, clone JES6-5H4, BD Biosciences, 554428), and tumor necrosis factor alpha (TNF-alpha) (rat Mab anti-mouse TNFa-FITC, clone MP6-XT22; 1/250 (BD Biosciences 554418) was performed for 30 min in the dark at 4 °C. Cells were washed with Perm/Wash and resuspended in PBS FACS and then analyzed by flow cytometry using the MACSQuant Analyzer. Live CD3+CD8+CD4− and CD3+CD8−CD4+ cells were gated and presented on a dot-plot. Two regions were defined to gate for positive cells for each cytokine; numbers of events found in these gates were divided by the total number of events in the parental population to yield percentages of responding T cells. The percentage obtained in medium alone was considered as background and subtracted from the percentage obtained with peptide stimulations. The threshold of positivity was defined according to experiment background, i.e., the mean percentage of stained cells obtained for each group in medium-alone condition plus two standard deviations (SD). Only the percentage of cytokine representing at least 50 events was considered as positive.

### 2.6. ELISA

Eight synthetic peptides representing HBV genotypes A–H were synthesized [15]; their sequences are provided in Appendix A. Peptides A, B, and A + B contained additional cysteine residues at the N-terminus for coupling. Each peptide was dissolved in sterile ddH_2_O at a concentration of 1 mg/mL and was tested at a final concentration of 2 μg/mL with enzyme-linked immunosorbent assay (ELISA). *E. coli*-expressed preS was used in ELISA at a final concentration of 2 μg/mL.

Sera were collected at different time points and stored at −20 °C. HBsAg and HBeAg levels in mouse sera were measured with commercial ELISA kits, respectively (Bio-Rad, Hercules, CA, USA and Diasorin SA, Antony, France). HBsAg concentrations were calculated in international units per milliliter (IU/mL) by reference to a standard curve established with known concentrations of HBsAg (Bio-Rad). Serum HBeAg levels were determined in hundred-fold diluted sera. Concentrations were calculated by reference to a standard curve established with the Paul-Ehrlich-Institut standard and are expressed in PEI U/mL.

For determination of specific IgG_1_ and IgG_2a_, ELISA plates (Nunc MaxiSorp 96-well flat bottom, Thermo Fisher Scientific, Cat. 442404) were coated overnight at 4 °C with preS or preS-derived peptides (2 µg/mL in 100 mM sodium bicarbonate buffer, pH 9.6) and washed two times with PBS 0.05% Tween 20 (PBST). Residual binding sites were blocked for 5 h at room temperature with 2% BSA/PBST, then the plates were incubated overnight at 4 °C with sera diluted 1:100, 1:500, 1:1000, 1:1500, 1:2000, 1:4000, 1:6000, 1:8000, and 1:10,000 for IgG_1_ and 1:50, 1:250, 1:500, 1:1000, 1:1500, 1:2000, 1:4000, 1:6000, and 1:8000 for IgG_2a_ measurements in 0.5% BSA/PBST. After five times washing with PBST, rat anti-mouse IgG_1_ (BD, Cat. 553440) and IgG_2a_ (BD, Cat. 553387) were diluted 1:1000 in 0.5% BSA/PBST and applied to the plate; after 2 h incubation at room temperature and five times washing, plates were incubated for 1 h at room temperature with horseradish-peroxidase (HRP)-linked goat anti-rat IgG (GE Healthcare, Cat. NA931V) at a dilution of 1:2000. After five times washing, the reaction was developed by incubation with substrate solution: 1 mg/mL 2,2′-Azino-bis(3-ethylbenzothiazoline-6-sulfonic acid) di-ammonium salt (ABTS) (Cat. A1888, Sigma-Aldrich, Burlington, MA, USA) in 70 mM citrate-phosphate buffer containing 0.003% H_2_O_2_ (Sigma-Aldrich, Cat.H1009). Absorbance values were measured at wavelength 405 nm (reference wavelength 492 nm) on a Tecan Infinite F50 spectrophotometer (Tecan Trading AG, Männedorf, Switzerland). The endpoint titration cut-off was determined as 3 × (mean + SD) values in the sera of the AAV-Empty + Vehicle group at a dilution of 1:100 for IgG_1_ and 1:50 for IgG_2a_.

### 2.7. ALAT/ASAT Measurement

Alanine aminotransferase (ALAT) and aspartate aminotransferase (ASAT) activities in sera of AAV-HBV- and AAV-Empty-injected mice were determined by the Laboratoire de Biologie Vétérinaire Vebiotel (Paris, France). Results are expressed in IU/liter.

### 2.8. HBV DNA Titration

Virions were purified from mouse sera by using a DNA purification kit (QIAamp Blood Mini kit, Qiagen, Hilden, Germany) and quantified by quantitative PCR as previously described [23]. Serial dilutions of the payw1.2 plasmid containing 1.2 copies of the HBV genome were used as quantification standards. The threshold of detection was 1020 IU/mL.

### 2.9. Statistical Analysis

Data were expressed as means ± standard errors of the means (SEM). Data was assumed to have a non-Gaussian distribution. When comparing groups, a Kruskal–Wallis test followed by a Dunn post hoc test was performed. The Mann–Whitney U test was used for all comparisons of two data sets. Statistical analysis was carried out using GraphPad Prism 8 software (Graphpad, San Diego, CA, USA).

## 3. Results

### 3.1. VVX001 Induces a Robust preS-Specific IgG Response in HBV- and Mock-Transfected Mice

In order to study if vaccination with preS-based HBV vaccines can overcome immune tolerance during chronic HBV infection, we used a murine model of chronic HBV infection. C57BL/6 mice (6 mice per group, 6 groups) were transduced with 5 × 10^10^ vg of either an AAV-HBV or AAV-Empty vector (Figure 1). Six weeks later, five subcutaneous injections of 20 µg of VVX001, preS, or placebo were administered at three-week intervals. Blood samples were taken before transduction, 5 weeks after transduction for mice randomization, and before each injection of vaccines. Subcutaneous injections were well tolerated, and no adverse effects were observed even after multiple administrations of the compounds. Mice were sacrificed two weeks after the last injections, and another blood sample was collected. Livers were removed following PBS perfusion.

The time courses of preS-specific IgG_1_ and IgG_2a_ antibody responses are depicted in Figure 2. Only a very slight and transient increase in preS-specific IgG_1_ was visible shortly after AAV-HBV transduction in the AAV-HBV groups at time point W5 before the first VVX001/preS/vehicle immunization. All VVX001- or preS-immunized groups showed a strong development of preS-specific IgG_1_ and IgG_2a_ already three weeks after the first injection in comparison with vehicle-immunized mice (Figure 2). The IgG_1_ levels arose fast and kept high levels over time in all non-vehicle groups which was also true for IgG_2a_ levels in preS-immunized groups. Whereas there seemed to be no difference in IgG_1_ levels between preS- and VVX001-immunized groups, the latter group showed a significantly higher IgG_2a_ response in comparison to preS-immunized mice. Also, preS-specific IgG_2a_ titers were higher in VVX001-immunized mice than in only preS-immunized mice (Figure 2, Appendix A). Notably, the VVX001-immunized AAV-HBV group shows a tendency to have higher anti-preS IgG responses than the AAV-Empty groups.

### 3.2. PreS-Specific IgG Responses in HBV- and Mock-Transfected Mice Are Directed Against the NTCP-Binding Site and Cross-React with preS Genotypes

The analysis of IgG_1_ response to peptides comprising the NTCP binding site (peptide A, amino acids 1–29) and the accessory domain involved in the inhibition of infection (peptide B, amino acids 30–61) or both sites (A + B, the genotype A peptide, amino acids 13–51) revealed a comparable reaction to the crucial region of the preS N-terminus (A + B) in all treated groups (Figure 3A). The reactivity to peptide B was more pronounced for HBV-transduced mice treated with VVX001, whereas HBV-carrier mice treated with preS reacted best to peptide A. The reverse was observed in mock-transduced groups: The VVX001 group recognized peptide A better whereas the preS-treated group reacted mainly to peptide B with low reaction to peptide A. No reactivity to the peptides was observed for the vehicle-immunized mice. To study the cross-reactivity to the crucial epitopes of common HBV genotypes, we synthesized peptides comprising amino acids 13–51 of preS of genotypes A-H and performed ELISA. We observed that groups immunized with VVX001 (both HBV- and mock-infected) reacted basically to all peptides of genotypes A-H; the same applies to the AAV-Empty + preS group. All the HBV-carrier mice immunized with preS showed low reactivity to the peptide of genotype G; however, they exhibited a comparable recognition of peptides of genotypes A, B, C, D, E, F, and H like VVX001-immunized mice (Figure 3B). Vehicle-immunized mice showed no IgG reactivity to any of the genotype peptides.

### 3.3. VVX001 and preS Augment Envelope-Specific Antibody-Secreting B Cells in AAV-HBV and AAV-Empty Mice

A B-cell ELISPOT assay was used to measure the anti-HBV B-cell responses. The capacity of B cells to secrete specific anti-HBV antibodies following five injections of vaccines was monitored in mice two weeks after the last injection (Figure 1). Splenocytes were incubated in plates coated with MHB (i.e., preS2 + S), HBcAg, or preS. KLH and anti-IgG antibody were used as negative and positive controls, respectively. In wells coated with anti-IgG antibodies, more than 1000 total secreting B cells/10^6^ splenocytes were detected for each group of mice. In AAV-Empty-transduced mice, both preS and MHBs-specific B cells were found after immunization with preS or VVX001 proteins. By contrast, only preS- but no MHB-specific B cells were found in HBV-carrier mice. Furthermore, the AAV-HBV-transduced groups had a lower number of responder mice than the AAV-Empty-transduced groups (preS-immunized: two versus four mice; VVX001-immunized mice: two versus five mice; see Figure 4). A trend towards a lower number of antibody-secreting B cells was also observed in the HBV-carrier mice versus the AAV-Empty mice, but the differences were statistically not significant (Figure 4).

### 3.4. Vaccination of HBV-Carrier Mice with VVX001 or preS Increases Intrahepatic Lymphocyte Numbers

Intrahepatic lymphocytes (IHLs) were collected from the livers of the different mouse groups. In the groups of mice that received only vehicles, AAV-HBV-transduced mice exhibited a tendency to have reduced IHL in comparison to AAV-Empty mice. HBV-carrier mice immunized with preS displayed a higher number of IHLs compared to the vehicle-immunized mice (*p* = 0.009) (Figure 5).

The capacity of T cells to produce IFNγ, IL-2, and/or TNF-alpha was evaluated by ICS assays. Splenocytes and IHLs of immunized mice were stimulated in vitro with 3 pools of peptides from the HBc, MHBs, and preS proteins or left unstimulated in medium alone. The production of cytokine-producing CD4^+^ or CD8^+^ T cells was detected by triple intracellular staining assays. A minimum of 2500 CD8^+^ and 20000 CD4^+^ splenocytes and 3000 CD8^+^ and CD4^+^ IHLs were acquired for the analysis, respectively. No cytokine-secreting T cells were found two weeks after the last injection in the spleen and liver of immunized mice. These results were in accordance with the results observed in ELISpot assays, where only a few IFNγ-secreting T-cells were detected. 

### 3.5. Vaccination with VVX001 and preS Does Not Increase Liver Enzymes

To investigate if the immunizations or AAV-HBV transductions were associated with the occurrence of liver dysfunction, alanine (ALAT) and aspartate aminotransferase (ASAT) activities were determined at W0, W6, +3W, +6W, +9W, +12W, and +14W post AAV-HBV or AAV-Empty injection (Figure 6). Six weeks after transduction and before any immunization, the mean level of ALAT was 35.7 ± 1.6 and 35.0 ± 1.1 IU/L in AAV-HBV and AAV-Empty transduced mice, respectively. A slight increase in ALAT values was observed in the sera of mice in all groups at +12W. ALAT concentrations did not differ significantly between groups at any time points. No significant hepatic dysfunction, e.g., >4× upper limits of normal levels of ALAT concentrations, was noticed. A similar pattern was found for ASAT activity (Appendix A). Starting from the time point +9W, levels of ALAT were the lowest in the VVX001 group among the HBV carrier mice (Figure 6, left panel).

### 3.6. Vaccination with VVX001 and preS Slightly Reduces HBV DNA and HBsAg Six Weeks After the First Immunization but Does Not Have Relevant Effects on HBeAg

Figure 7 shows the expression of HBV (i.e., HBV DNA and HBeAg) in the AAV-HBV-transduced mice during the complete period of the experiments. In fact, persistence of HBV for up to 50 weeks has been reported for the mouse model used [21]. Before immunization, the mean level of HBV DNA was 5.77 ± 0.15 log_10_ IU/mL (5.87 ± 0.24, 5.86 ± 0.31, and 5.5 ± 0.16 log_10_ IU/mL in groups immunized with VVX001, preS, and vehicle, respectively). HBV DNA levels decreased 6 weeks after the first injection, especially in the group of mice immunized with preS but also in VVX001-immunized mice (Figure 7, upper left panel).

The range of HBeAg concentrations before immunization was 18 to 156 PEI U/mL. HBeAg levels were calculated as the percentage of HBeAg decrease compared to pre-immunization levels (Figure 7). No significant changes in HBeAg levels were observed during the follow-up in mice receiving VVX001, preS, or vehicle.

The HBV-carrier mice were allocated into groups with the same average HBsAg level before receiving the vaccine. HBsAg mean levels were 12,103 ± 4730, 14,419 ± 5127, and 15,788 ± 7009 IU/mL in AAV-HBV mouse groups immunized with VVX001, preS, and vehicle, respectively (Figure 8, left panel). The levels of HBsAg started to decrease three weeks after immunization. To better appreciate the effect of immunization, HBsAg levels were calculated as the percentage of HBsAg decrease compared to pre-immunization levels (Figure 8, right panel). Six weeks after immunization, the vehicle group had returned to baseline levels, while the groups receiving either preS or VVX001 injections showed a decline in serum HBsAg (Figure 8, right panel). HBsAg levels remained reduced in the preS-immunized groups after the third injection (+9W). 

## 4. Discussion

We have previously shown that the recombinant grass pollen allergy vaccine BM32, which is composed of four recombinant fusion proteins consisting of HBV-derived preS as a carrier and peptides from the four major grass pollen allergens, Phl p 1, Phl p 2, Phl p 5, and Phl p 6, induced preS-specific IgG antibodies in grass pollen allergic patients [12,15]. We found that the preS-specific IgG antibodies induced by vaccination with BM32 were able to neutralize in vitro HBV infection because the BM32-induced antibodies were directed against the NTCP-binding site of preS and thus blocked HBV infection [12]. Recently, we could show that a single component of BM32, i.e., BM325, termed VVX001, was able to induce such preS-specific HBV-neutralizing antibodies in an individual who was non-responsive to HBsAg-based HBV vaccines [14].

We hypothesized that VVX001 may not only be useful for inducing HBV-neutralizing antibodies in non-responders to HBsAg-based vaccines but might also be a candidate vaccine for therapeutic vaccination in CHB. This hypothesis is based on the idea that VVX001 can induce preS-specific antibodies which may interrupt recurrent infection of liver cells in the course of CHB and additionally generate “mild” preS-specific CD4+ and CD8+ T-cell responses as was observed in grass pollen allergic patients vaccinated with BM32 [12]. Such a “mild T cell response” may in the long term contribute to the elimination of infected liver cells without causing severe liver damage. It was a major goal of this study to investigate if VVX001 can induce a preS-specific antibody response in an in vivo model of CHB because we had observed that patients with chronic HBV infections mounted no relevant preS-specific antibody responses [12]. This lack of preS-specific antibody response in CHB may be due to low immunogenicity of preS because it is expressed/present on circulating HBV or on HBV-derived subviral particles in low amounts and/or due to immunological tolerance in the host. We therefore immunized AAV-HBV mice, which are transduced with an AAV carrying a replication-competent HBV-DNA genome, allowing HBV to replicate in the liver [21]. Like patients with CHB, these mice did not spontaneously develop relevant levels of preS-specific antibodies, as can be observed in Figure 2 for the AAV-HBV + vehicle-immunized mice through the complete observation period and for AAV-HBV mice before vaccination with preS-containing vaccines five weeks after transduction. However, a strong increase in preS-specific IgG antibodies was observed for AAV-HBV as well as for AAV-Empty mice, demonstrating that the presence of HBV did not prevent the development of a robust preS-specific IgG response. Although the difference was statistically not significant for IgG_1_, it is noteworthy that IgG_2a_ antibody levels in HBV-carrier mice immunized with VVX001 were significantly higher than in mice immunized with preS alone (Appendix A) and mock-transduced mice. Furthermore, preS-specific IgG_2a_ titers were higher in VVX001-immunized mice than in only preS-immunized mice (Figure 2, Appendix A).

Importantly, preS-specific IgG antibodies induced in HBV-carrier mice were directed against the epitope represented by the preS-derived peptides A and B, which contain the NTCP binding site of preS. One can therefore bona fide assume that these antibodies will have HBV neutralizing activity. However, due to lack of serum, we could not study the in vitro HBV-neutralizing effects of the mouse sera. It is another limitation of our study that the therapeutic effects of the preS-specific antibodies could not be investigated in the AAV-HBV mouse model because in this model the HBV-DNA and HBsAg levels are not a result of chronic infection but are due to AAV transduction of liver cells producing constantly HBV-DNA and HBV antigens [24,25]. Furthermore, the AAV-HBV mice lack human NTCP on their liver cells so that re-infection via this receptor cannot occur [26]. Therefore, therapeutic vaccination with VVX001 showed no significant effect on the murine model of chronic HBV. However, we were able to demonstrate that preS-specific antibodies induced with VVX001 containing preS from genotype A cross-reacted with the corresponding NTCP-binding site-containing peptides from genotypes B, C, D, E, F, G, and H in AAV-HBV mice. One may therefore hope that vaccination VVX001 may convey cross-protection for other HBV genotypes. The development of preS-specific antibodies was accompanied by the formation of B cells/plasmablasts secreting preS-specific antibodies, as demonstrated by ELISPOT analysis, indicating that a corresponding secondary preS-specific B-cell response was generated by immunization with VVX001. Furthermore, like in three other studies performed in the AAV-HBV model or in similar models with vaccines intended for therapeutic vaccination, we found evidence for the development of vaccine-specific T-cell responses [27,28,29]. In fact, HBV-carrier mice vaccinated with VVX001 or preS showed increased intrahepatic lymphocytes, suggesting that cells activated by the vaccine could be recruited or attracted into the liver by the HBV-expressing hepatocytes. Vaccination with VVX001 or preS was safe because no adverse effects were observed and ALAT and ASAT levels remained in the normal range, indicating that the vaccines are well tolerated.

Moderate favorable effects on HBV DNA and HBsAg levels like for other therapeutic vaccines tested in the AAV-HBV or in comparable murine models were observed [27,28,29]. However, as indicated above, the major mechanism of VVX001, i.e., the blocking of recurrent HBV liver infection could not be tested in the AAV model because the mice do not express human NTCP on their liver cells. Nevertheless, our study has clearly demonstrated that immunization with VVX001 can induce preS-specific antibodies in a murine model of CHB. Our vaccine is a simple subunit vaccine formulated with aluminum hydroxide, an adjuvant which has been used in numerous vaccines with high safety profile. Subunit vaccine like VVX001 have the advantage, that they can be well dosed and induce distinct but mild T-cell responses avoiding excessive and eventually cytotoxic T-cell responses which may be important for the vaccination of patients with CHB who may be at risk for side effects. Our study therefore encourages further evaluation of VVX001 for therapeutic vaccination against CHB.

## Figures and Tables

**Figure 1 vaccines-13-00854-f001:**
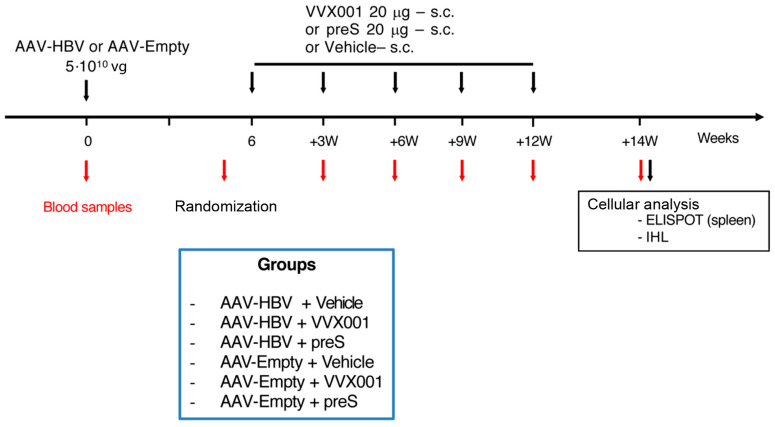
Scheme of treatment in a murine model of CHB. Groups of 6–8-week-old C57BL/6 mice (n = 6) were injected (black arrows) subcutaneously with 20 μg of VVX001, preS or vehicle (aluminum hydroxide) 6 weeks (W) after AAV-HBV or AAV-Empty transduction. Four additional injections were performed every 3 weeks. Blood samples (red arrows) were taken during the study as indicated. Mice were sacrificed 14 weeks after the first vaccination and immunological analyses were performed. AAV, adeno-associated virus. ELISPOT, enzyme-linked ImmunoSpot. IHL, intrahepatic lymphocytes. vg, viral genome equivalents.

**Figure 2 vaccines-13-00854-f002:**
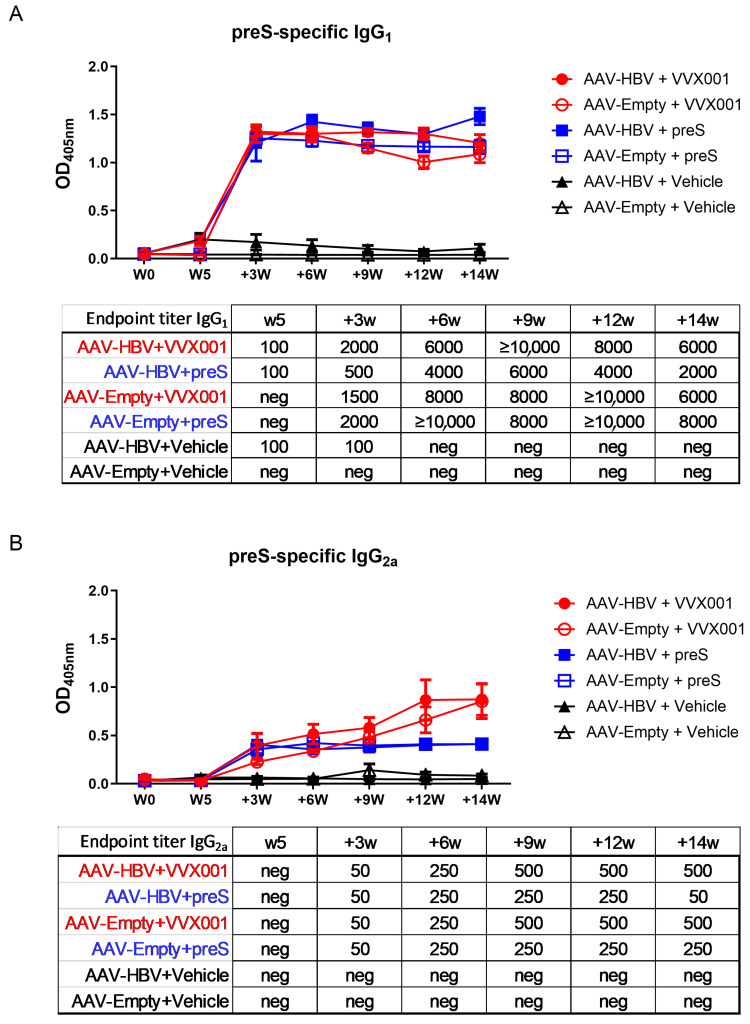
VVX001 induces preS-specific IgG_1_ and IgG_2a_ responses in immunized mice. Shown are the endpoint titers (tables) and optical density (OD) values (y-axes) corresponding to (**A**) preS-specific IgG_1_ (dilution 1:100) and (**B**) IgG_2a_ (dilution 1:50) in mice immunized with five injections of VVX001, preS, or placebo as time course (x-axes). Results are depicted as mean values ± SEM.

**Figure 3 vaccines-13-00854-f003:**
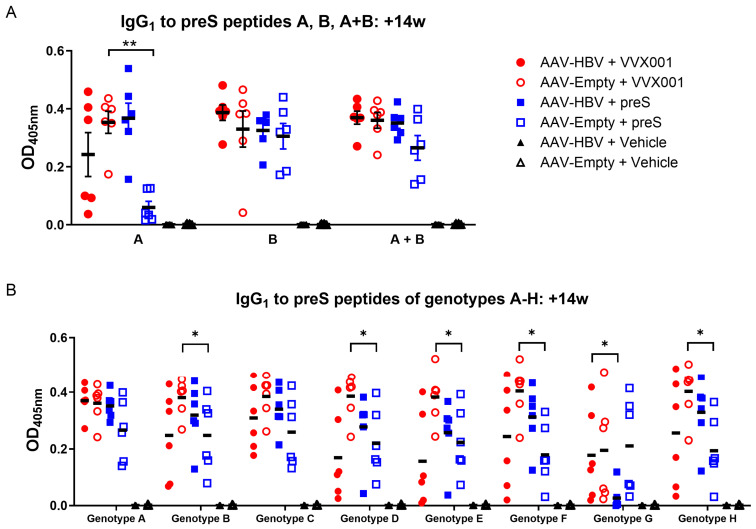
Vaccination with VVX001 induces IgG antibodies recognizing the preS receptor binding domain of HBV genotypes A–H. Shown are optical density (OD) values (y-axes) corresponding to IgG_1_ antibody levels specific for preS-derived peptides (**A**) A, B, and A + B (i.e., genotype A peptide) comprising the NTCP binding site and the accessory domain involved in the inhibition of infection and (**B**) for peptides of genotypes A, B, C, D, E, F, G, and H in mice immunized with VVX001, preS or Vehicle at the time point +14W expressed as mean values. Significant differences between the immunized groups are indicated: * *p* < 0.05, ** *p* < 0.01.

**Figure 4 vaccines-13-00854-f004:**
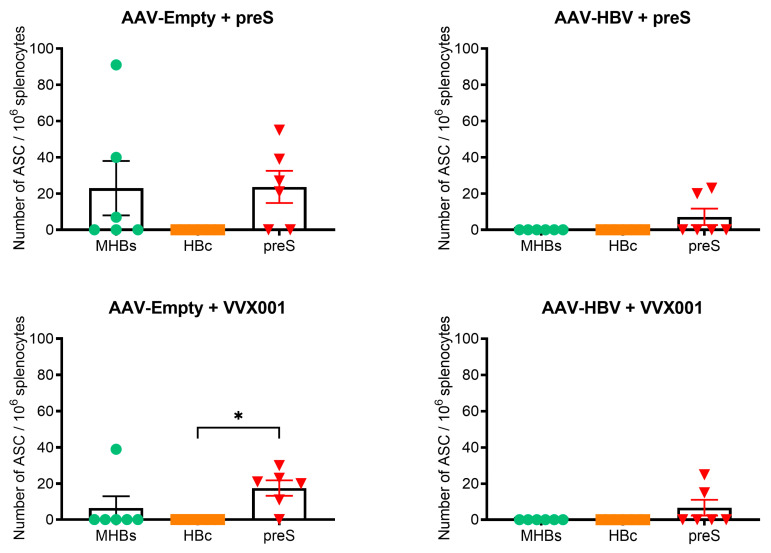
VVX001 and preS induce envelope-specific antibody-secreting B-cell responses. Two weeks after the last injection, HBV-specific B-cell response was tested by B-cell ELISPOT assay. Results are expressed as number of antibody-secreting B cells per million splenocytes. Shown is ex vivo quantification of MHBs- (green), HBc- (orange), preS-specific (red) antibody-secreting B cells. Individual responses in groups of AAV-Empty or AAV-HBV transduced mice immunized with preS (**upper panels**) and VVX001 (**lower panels**). Each symbol stands for one mouse. Data are shown as means ± SEM. Significant differences are indicated: * *p* < 0.05.

**Figure 5 vaccines-13-00854-f005:**
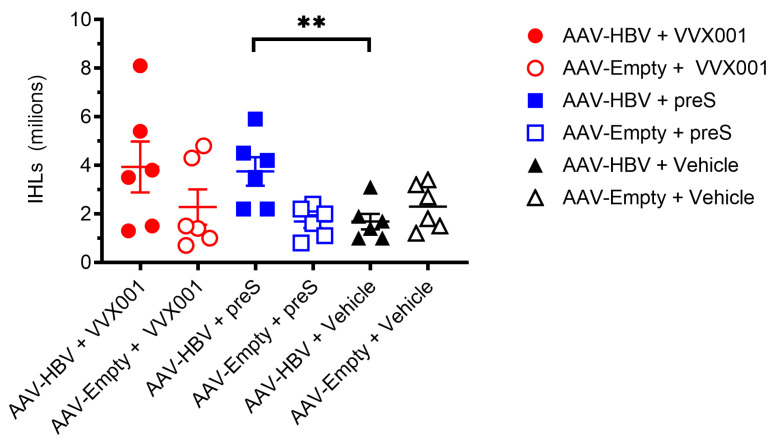
VVX001 and preS-immunized HBV-carrier mice have higher numbers of intrahepatic lymphocytes than non-vaccinated animals. Two weeks after the last injection (time point +14w) IHLs were harvested and counted (y-axis). Each symbol stands for one mouse. Indicated are the mean numbers of million cells. Significant differences are indicated: ** *p* < 0.01.

**Figure 6 vaccines-13-00854-f006:**
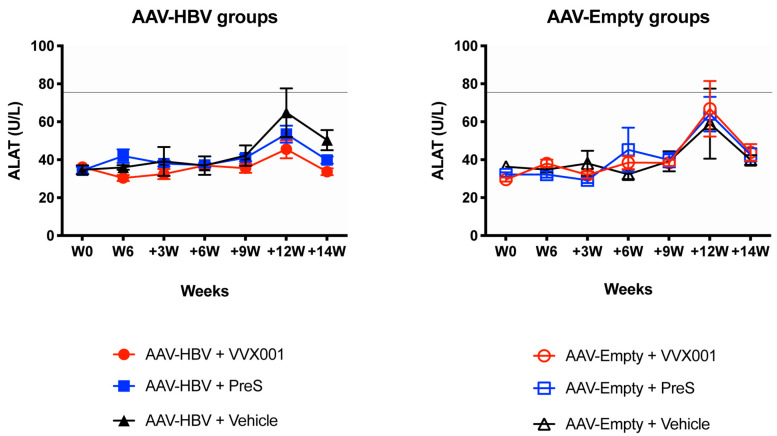
Alanine aminotransferase activity during the experiments. ALAT activity was measured in AAV-HBV (**left panel**) and in AAV-Empty (**right panel**) groups. ALAT levels are given in international units per liter (IU/L) and are expressed as mean values ± SEM, a horizontal line indicates the upper limit of the normal ALAT level.

**Figure 7 vaccines-13-00854-f007:**
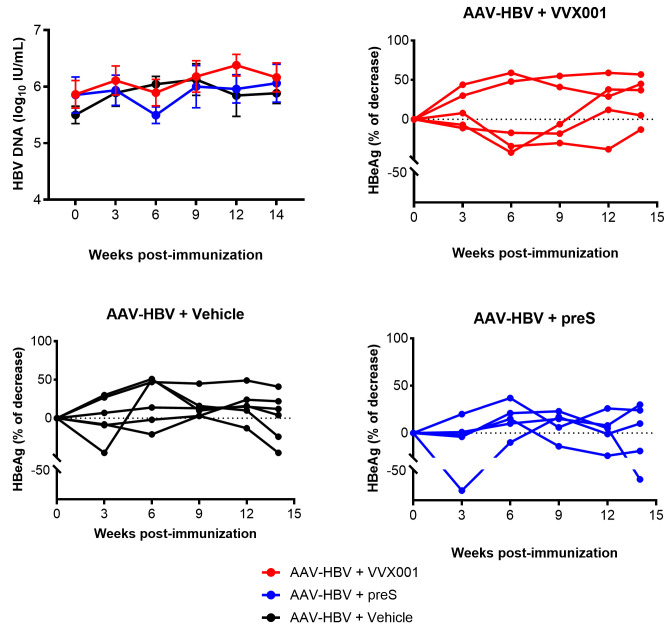
Immunization with preS-containing vaccine reduces HBV DNA levels 6 weeks post immunization but does not induce significant changes in serum HBeAg levels. Blood samples were taken at various times after AAV-HBV transduction and HBV DNA (**upper left panel**) and HBeAg levels (**upper right**, **lower left** and **right panels**) are shown (y-axes) for AAV-HBV transduced mice with different treatments (VVX001, preS, or vehicle). Individual levels of HBeAg in the sera of VVX001- (red lines), preS- (blue lines) and Vehicle-immunized (black lines) mice are expressed as % of decrease compared to HBeAg pre-vaccination levels. HBV DNA levels are expressed as mean values ± SEM in international units/mL (log_10_).

**Figure 8 vaccines-13-00854-f008:**
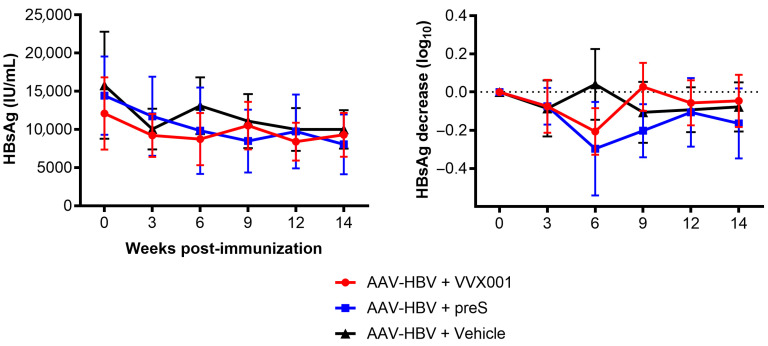
Follow-up of HBsAg levels in sera from AAV-HBV transduced mice after immunization with different vaccines or vehicle. HBsAg levels are expressed as mean values ± SEM in HBsAg international units/mL (**left panel**) and as changes in HBsAg levels (log10 of HBsAg concentration) compared to HBsAg pre-vaccination levels (**right panel**) following administration of compounds according to the groups of immunization.

## Data Availability

All the data supporting the findings of this study are available in the text, figures, tables, Appendix A, or from the corresponding author upon reasonable request.

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
