# Peer review of "VVX001 Induces preS-Specific Antibodies Reacting to Common HBV Genotypes in Hepatitis B Virus (HBV) Carrier Mice"

_vaccines, 2025, doi:10.3390/vaccines13080854_

Round 1

Reviewer 1 Report

Comments and Suggestions for Authors

It is a very sophisticate study that suggest a potential role of a candidate for therapeutic vaccine against chronic HBV infections. The results are not strong but can be a good track to be followed.

Obs.: At line 392, there is a missing "t" in "stumulated"

Author Response

Comment 1: It is a very sophisticate study that suggest a potential role of a candidate for therapeutic vaccine against chronic HBV infections. The results are not strong but can be a good track to be followed.

Reply 1: We thank the reviewer for the comment. Indeed, the study has been a first step towards a clinical study (see: NCT03625934).

Comment 2: Obs.: At line 392, there is a missing "t" in "stumulated"

Reply 2: Corrected as requested (see line 407 of revised manuscript).

Reviewer 2 Report

Comments and Suggestions for Authors

Using a mouse model, the authors evaluated the ability to induce an immune response during chronic infection with hepatitis B. To model chronic infection, the authors used transduction with an adeno-associated virus carrying replication competent HBV-DNA genome. The authors showed that in this model, administration of their vaccine VVX001, carrying the preS domain of the large surface protein of hepatitis B virus, is indeed able to induce the formation of specific antibodies and attract lymphocytes to the liver. At the same time, no toxic effects on the liver were detected.

Overall, all results are clearly presented and well illustrated. All conclusions are reasonable.

There are comments to the article that need to be corrected:

  1. Lines 105-108 It should be corrected that not only adenoviral vectors but also adeno-associated viral vectors were used. In reference 19 an adeno-associated virus was used and in reference 20 an adenovirus of serotype 5 was used.
  2. It is recommended to clarify whether there was indeed long-term expression of HBV in all AAV-HBV transduced mice.
  3. Line 329-333 Statistical validity should be cited for these statements (it is recommended to indicate on the graph).
  4. Figures 3-5 should indicate statistical differences between the groups discussed in the text.
  5. 5.Line 477-478 “we had observed that patients with chronic HBV infections mounted no relevant preS-specific antibody responses” - the reference is required

Author Response

Comment 1: Lines 105-108 It should be corrected that not only adenoviral vectors but also adeno-associated viral vectors were used. In reference 19 an adeno-associated virus was used and in reference 20 an adenovirus of serotype 5 was used.

Reply 1: Following the reviewers request, the sentence was corrected (see lines 105-107 of revised and marked manuscript, please see the attachment).

Comment 2: It is recommended to clarify whether there was indeed long-term expression of HBV in all AAV-HBV transduced mice.

Reply 2: The results presented in Figure 7 and 8 support the long-term expression of HBV in all AAV-HBV-transduced mice over the observation period.  Furthermore, persistence of up to 50 weeks of HBV was described for this model (Figure 1 and Figure 3 of Dion S, Bourgine M, Godon O, Levillayer F, Michel M 2013. Adeno-Associated Virus-Mediated Gene Transfer Leads to Persistent Hepatitis B Virus Replication in Mice Expressing HLA-A2 and HLA-DR1 Molecules. J Virol 87:. https://doi.org/10.1128/jvi.03134-12 ). This has been mentioned in the revised manuscript (see lines 436-438).  

Comment 3: Line 329-333 Statistical validity should be cited for these statements (it is recommended to indicate on the graph).

Reply 3: Statistically significant differences of antibody responses are indicated in revised Figure 3. We show the differences only between actively immunized groups. Differences between non-immunized and immunized mice were always significant and therefore were not indicated. There were no relevant antibody responses in non-vaccinated mice. Therefore also AAV-HBV and AAV-Empty groups without vaccination were not compared with each other.

Comment 4: Figures 3-5 should indicate statistical differences between the groups discussed in the text.

Reply 4: The statistically significant differences were indicated on Figures 3-5.

Comment 5: Line 477-478 “we had observed that patients with chronic HBV infections mounted no relevant preS-specific antibody responses” - the reference is required

Reply 5: As requested by the reviewer, the quoted reference was added (Immunotherapy With the PreS-based Grass Pollen Allergy Vaccine BM32 Induces Antibody Responses Protecting Against Hepatitis B Infection. Cornelius C, Schöneweis K, Georgi F, Weber M, Niederberger V, Zieglmayer P, Niespodziana K, Trauner M, Hofer H, Urban S, Valenta R. EBioMedicine. 2016 Sep;11:58-67. doi: 10.1016/j.ebiom.2016.07.023. Epub 2016 Aug 8).

Reviewer 3 Report

Comments and Suggestions for Authors

This study analyzed the immune response after VVX001 immunization in the AAV-HBV murine model. In the AAV murine model, VVX001 immunization successfully induced antibodies against preS. On the other hand, there is no significant difference between VVX001 and preS immunization. And there is no significant decrease in levels of HBV-DNA or HBs antigen with VVX001 immunization. Therefore, I cannot find the significance of using VVX001 as a therapeutic vaccine.

Major comments

  1. As indicated, both VVX001 and preS immunization successfully induced antibodies against preS in the AAV-HBV murine model. Moreover, there is no significant difference between the two in terms of immunological outcomes. Additionally, there is no notable reduction in HBV-DNA or HBs antigen levels with VVX001 immunization. Based on these findings, it is difficult to establish the added value or therapeutic significance of VVX001 over preS immunization.

  1. Neutralizing activity of antibodies should be evaluated using primary human hepatocytes. Evaluating neutralization in human hepatocytes would provide more clinically relevant data regarding the potential efficacy of VVX001 as therapeutic vaccine.

  1. In reference to Figure 2, it is important to clarify whether the antibody levels have reached a plateau. The data should be presented using endpoint titers rather than OD values, as OD may not accurately reflect the true antibody concentration.

Author Response

Comment 1: As indicated, both VVX001 and preS immunization successfully induced antibodies against preS in the AAV-HBV murine model. Moreover, there is no significant difference between the two in terms of immunological outcomes. Additionally, there is no notable reduction in HBV-DNA or HBs antigen levels with VVX001 immunization. Based on these findings, it is difficult to establish the added value or therapeutic significance of VVX001 over preS immunization.

Reply 1: We have analyzed preS-specific antibodies in greater detail performing also titration experiments as requested by the reviewer. Results obtained in the newly added Table S2 show that VVX001 induces significantly higher preS-specific IgG2a antibody responses as compared to preS and also endpoint titers for IgG2a are higher. See newly added Table S3 and revised Figure 2 and lines 324-328 and 508-513 of the revised and marked manuscript (please see the attachment).

Regarding HBV-DNA and HBs antigen levels we would like to emphasize that in the mouse model used the HBV-DNA and HBs antigen levels are not a result of chronic infection but are due to adenoviral transduction of liver cells producing constantly HBV-DNA and HBs antigen. Therefore, unlike in humans reinfection of liver cells cannot occur in the mouse model because mouse liver cells are not susceptible to reinfection. Accordingly the mouse model can only be used to investigate if one can establish a preS-specific immune response in mice containing large amounts of HBV antigen. Our results show that the latter objective was achieved. In fact we could show the induction of preS-specific antibody response in presence of high viremia and we found absence of acute liver inflammation indicating safety of the vaccination. We have revised our manuscript to highlight these important points and the limitations of the mouse model used (see lines xxx of revised manuscript). Furthermore we have acknowledged the limitation that the mouse model cannot be used to study therapeutic effects. Accordingly we have revised the title of our manuscript to highlight the major finding, i.e., that our vaccine can induce preS-specific antibodies in the model.    

Comment 2: Neutralizing activity of antibodies should be evaluated using primary human hepatocytes. Evaluating neutralization in human hepatocytes would provide more clinically relevant data regarding the potential efficacy of VVX001 as therapeutic vaccine.

Reply 2: We were limited regarding volumes of mouse sera but we have shown the neutralizing activity earlier for patients and rabbit antisera which showed the very same epitope specificity as the mouse sera in our study (Immunotherapy With the PreS-based Grass Pollen Allergy Vaccine BM32 Induces Antibody Responses Protecting Against Hepatitis B Infection. Cornelius C, Schöneweis K, Georgi F, Weber M, Niederberger V, Zieglmayer P, Niespodziana K, Trauner M, Hofer H, Urban S, Valenta R. EBioMedicine. 2016 Sep;11:58-67. doi: 10.1016/j.ebiom.2016.07.023. Epub 2016 Aug 8). Accordingly, one may assume that the antibodies induced in the mouse model will have neutralizing properties. This was mentioned in the revised discussion (see lines 481-486 of the revised discussion). Furthermore, we mentioned that we did not have enough mouse sera for neutralization as a limitation of our study (lines 514-518 of the revised discussion).

Comment 3: In reference to Figure 2, it is important to clarify whether the antibody levels have reached a plateau. The data should be presented using endpoint titers rather than OD values, as OD may not accurately reflect the true antibody concentration.

Reply 3: We thank the reviewer for this excellent suggestion and performed the requested experiments. The endpoint titrations were done for IgG1 and IgG2a see revised Figure 2 and newly added Table S3 of the revised manuscript. Results were mentioned in the revised manuscript (lines 324-328).  

Reviewer 4 Report

Comments and Suggestions for Authors

In this manuscript, Tulaeva and colleagues tested a protein vaccine, VVX001, which is a fusion between PreS domain of HBsAg and grass pollen allergen peptides, on a mouse model of chronic hepatitis B using AAV-based system. Five doses of VVX001 20microgram was immunized after AAV-HBV transduction and establishment of viral replication in C57 mice. The authors demonstrated the induction of preS-specific IgGs and their respective antibody-producing B cells although the preS control did the same. However, compared with preS or vehicle controls, no significant changes in HBeAg, and  HBsAg titre were observed during 14 weeks following the vaccinations.

Overall, this project was reasonably devised. The only caveat in this system lies in the lack of understanding of the AAV-HBV system. The HBV genome is introduced into the mice liver with the help of AAV particles, and viral replication can be initiated. However, HBV cannot spread among the hepatocytes due to the lack of a functional receptor, which simply nullify any therapeutic effect of anti-PreS antibodies that solely targets viral entry.

Overall, the data of this manuscript is sound, but the interpretations are highly biased. I am deeply concerned by the authors’ conclusion of “VVX001 breaks immune tolerance”  despite the data clearly indicated otherwise.

As such, this manuscript is acceptable only when the authors fully acknowledge the lack of effect of VVX001 on a murine model of chronic hepatitis B.

Author Response

Comment 1: In this manuscript, Tulaeva and colleagues tested a protein vaccine, VVX001, which is a fusion between PreS domain of HBsAg and grass pollen allergen peptides, on a mouse model of chronic hepatitis B using AAV-based system. Five doses of VVX001 20microgram was immunized after AAV-HBV transduction and establishment of viral replication in C57 mice. The authors demonstrated the induction of preS-specific IgGs and their respective antibody-producing B cells although the preS control did the same. However, compared with preS or vehicle controls, no significant changes in HBeAg, and  HBsAg titre were observed during 14 weeks following the vaccinations.

Overall, this project was reasonably devised. The only caveat in this system lies in the lack of understanding of the AAV-HBV system. The HBV genome is introduced into the mice liver with the help of AAV particles, and viral replication can be initiated. However, HBV cannot spread among the hepatocytes due to the lack of a functional receptor, which simply nullify any therapeutic effect of anti-PreS antibodies that solely targets viral entry.

Reply 1: We thank the reviewer for this comment. As indicated in our reply to another reviewer, we have analyzed preS-specific antibodies in greater detail performing also titration experiments as requested by this other reviewer. Results obtained in the newly added Table S2 show that VVX001 induces significantly higher preS-specific IgG2a antibody responses as compared to preS and also endpoint titers for IgG2a are higher. See newly added Table S3 and revised Figure 2 and lines 324-328 and 508-513 of the revised and marked manuscript (please see the attachment).

Regarding HBV-DNA and HBV antigens levels we fully agree with this reviewer. Indeed, in the mouse model used the HBV-DNA and HBsAg levels are not a result of chronic infection but are due to adenoviral transduction of liver cells producing constantly HBV-DNA and HBsAg. Therefore, unlike in humans, reinfection of liver cells cannot occur in the mouse model because mouse liver cells are not susceptible to reinfection. Accordingly, the mouse model can only be used to investigate if one can establish a preS-specific immune response in mice containing large amounts of HBV antigens. Our results show that the latter objective was achieved. In fact, we could show the induction of preS-specific antibody response in presence of “high viremia” and we found absence of acute liver inflammation indicating safety of the vaccination. We have revised our manuscript to highlight these important points and the limitations of the mouse model used (see lines 25-27, 514-537 and 544-546 of the revised manuscript). Furthermore, we have acknowledged the limitation that the mouse model cannot be used to study therapeutic effects. Accordingly we have revised the title of our manuscript to highlight the major finding, i.e., that our vaccine can induce preS-specific antibodies in the model. 

Comment 2: Overall, the data of this manuscript is sound, but the interpretations are highly biased. I am deeply concerned by the authors’ conclusion of “VVX001 breaks immune tolerance”  despite the data clearly indicated otherwise.

Reply 2: We agree with the reviewer and revised the title to emphasize the major finding, i.e., that our vaccine can induce preS-specific antibodies in the murine model of chronic hepatitis B. 

Comment 3: As such, this manuscript is acceptable only when the authors fully acknowledge the lack of effect of VVX001 on a murine model of chronic hepatitis B.

Reply 3: Following the reviewers request we have fully acknowledged that in the murine model of chronic hepatitis used vaccination with VVX001 lacked effect (see lines 518-525 of the revised discussion). 

Round 2

Reviewer 3 Report

Comments and Suggestions for Authors

Thanks to the authors for revising the manuscript. The manuscript was improved.

Author Response

Comment of the Reviewer 3: Thanks to the authors for revising the manuscript. The manuscript was improved.

Reply: We thank the reviewer for this comment.

Reviewer 4 Report

Comments and Suggestions for Authors

I have no further comments on the revised manuscript except for one factual error. The HBV genome is introduced via the adenovirus-associated virus vector as opposed to Adenovirus. 

Author Response

Comment of the Reviewer 4: I have no further comments on the revised manuscript except for one factual error. The HBV genome is introduced via the adenovirus-associated virus vector as opposed to Adenovirus

Reply: We thank the reviewer for the careful revision of the manuscript. We have corrected this mistake (line 528 of the revised marked manuscript or line 522 of the revised clean manuscript).